# Dual Cannabinoid and Orexin Regulation of Anhedonic Behaviour Caused by Prolonged Restraint Stress

**DOI:** 10.3390/brainsci13020314

**Published:** 2023-02-13

**Authors:** Hye Ji J. Kim, Ayat Zagzoog, Costanza Ceni, Rebecca Ferrisi, Nicola Janz, Robert B. Laprairie

**Affiliations:** 1College of Pharmacy and Nutrition, University of Saskatchewan, Saskatoon, SK S7N 5E5, Canada; 2Department of Pharmacy, University of Pisa, 56126 Pisa, Italy; 3Doctoral School in Life Sciences, University of Siena, 53100 Siena, Italy; 4Department of Pharmacology, College of Medicine, Dalhousie University, Halifax, NS B3H 4R2, Canada

**Keywords:** anhedonia, restraint stress, endocannabinoid system, cannabinoid, orexin

## Abstract

The endocannabinoid and orexin systems share many biological functions, including wakefulness, stress response, reward processing, and mood. While these systems work against one another with respect to arousal, chronic stress-induced downregulation of both systems often leads to anhedonia or the inability to experience pleasure from natural rewards. In the current study, a 24 h restraint stress test (24 h RST) reduced sucrose preference in adult male and female C57BL/6 mice. Prior to the stressor, subsets of mice were intraperitoneally administered cannabinoid and orexin receptor agonists, antagonists, and combinations of these drugs. Restraint mice that received the cannabinoid receptor type 1 (CB1R) antagonist SR141716A, orexin receptor type 2 (OX2R) agonist YNT-185, and the combination of SR141716A and YNT-185, exhibited less anhedonia compared to vehicle/control mice. Thus, the 24 h RST likely decreased orexin signaling, which was then restored by YNT-185. Receptor colocalization analysis throughout mesocorticolimbic brain regions revealed increased CB1R-OX1R colocalization from SR141716A and YNT-185 treatments. Although a previous study from our group showed additive cataleptic effects between CP55,940 and the dual orexin receptor antagonist (TCS-1102), the opposite combination of pharmacological agents proved additive for sucrose preference. Taken together, these results reveal more of the complex interactions between the endocannabinoid and orexin systems.

## 1. Introduction

Anhedonia is a core symptom in mood disorders such as major depression [1,2] and anxiety [3] and is described as the inability to feel pleasure from natural, non-drug sources [1]. As a form of dopaminergic dysregulation, anhedonia may also appear in substance use disorders (SUD) [4]. The mesocorticolimbic circuit that regulates hedonic and anhedonic behaviours involves dopaminergic projections from the ventral tegmental area (VTA) to subcortical regions including the nucleus accumbens (NAC), medial prefrontal cortex (MPFC), and cingulate cortex [5,6]. These brain regions process wanting, liking, and learning with respect to rewarding stimuli [7,8]. Anhedonia may emerge in times of stress, as wanting or choosing to consume rewarding drugs over natural rewards is a form of coping in humans [4,9,10]. Preclinically, both chronic and acute patterns of homo- and/or heterotypic physiological and psychological stress produce anhedonia [11,12,13].

The endocannabinoid and orexin systems are among the many neuromodulatory systems involved in anhedonia [14,15]. The endocannabinoid system (ECS) includes 2 G-protein coupled receptors (GPCRs), the type 1 (CB1R) and type 2 (CB2R) cannabinoid receptors [16]. These receptors are activated by the endogenous ligands, 2-arachidonoylglycerol (2-AG) and anandamide (AEA) [17]. CB1R and CB2R are expressed throughout the central nervous system and modulate neuropsychological and homeostatic functions like sedation, body temperature, nociception, and appetite [18]. Within the orexin system, endogenous orexin A (OXA) and B (OXB) activate orexin receptor type 1 (OX1R) and type 2 (OX2R), both of which are also GPCRs [19]. Orexin signaling leads to arousal [20], hyperthermia [21], energy expenditure related to movement, and increased appetite [22,23]. Both endocannabinoid and orexin systems regulate reward [24] and emotionality [25,26,27]. Whereas the dopamine system is believed to drive “wanting,” the ECS is thought to determine “liking” the rewarding stimuli [28,29]. Moreover, CB1R activation following stress is associated with antidepressant effects [30,31]. Orexin receptor antagonists, on the other hand, are anxiolytic and inhibit stress cue-associated hyperarousal in rodent models of post-traumatic stress disorder (PTSD) [32,33].

The shared functions of these neuromodulatory systems stem from receptor co-localization and possible heterodimerization in the cerebral cortex, hippocampus, thalamus, hypothalamus, amygdala, VTA, periaqueductal gray, dorsal raphe nucleus, and cerebellum [34,35,36,37,38,39]. Bioluminescence and fluorescence resonance energy transfer experiments demonstrate that CB1R forms physical interactions with both orexin receptor subtypes [37], and CB1R agonist-mediated receptor internalization is associated with the presence of CB1R-OX1R heterodimer-filled vesicles [35,36]. Activation of one receptor type also triggers intracellular signaling and endogenous ligand production of the other [34,37]. In a previous publication by our group, co-administration of the cannabinoid receptor agonist CP55,940 and dual orexin receptor antagonist TCS-1102 was associated with additive cataleptic effects along with increased CB1R-OX1R colocalization in the ventral striatum [40].

The aim of the current study was to expand our knowledge of cannabinoid–orexin receptor functionality in neuropsychiatric behaviours. A 24 h restraint stress test (RST) produced anhedonic behaviour in mice [12]. As both systems mediate hedonic feeding and drug-induced reward [15], combinations of cannabinoid and orexin receptor agonists and antagonists were tested for their ability to prevent prolonged stress-induced anhedonia in the sucrose preference test (SPT) [41]. The mesocorticolimbic circuit was then evaluated by confocal microscopy, which provided additional insights into brain region-specific cannabinoid–orexin receptor interactions.

## 2. Materials and Methods

### 2.1. Compounds

CP55,940 (cannabinoid receptor agonist, Cat #90084, Batch #0541376-1), TCS-1102 (dual orexin receptor antagonist [DORA], Cat #18495, Batch #0526753-6), and SR141716A (CB1R antagonist, Cat #9000484, Batch #0469387-42) were sourced from Cayman Chemical Company (Ann Arbor, MI, USA). YNT-185 (OX2R agonist) was purchased from Tocris Bioscience (Bristol, United Kingdom, Cat #1804978-82-2, Batches #1A/259153 and A/270184). All compounds were stored at −20 °C. CP55,940 was first dissolved in 100% methanol before being added to a vehicle solution designated for mice consisting of 1:1:18 of 100% ethanol, Kolliphor EL (MilliporeSigma, Oakville, ON, Canada), and 1 M phosphate-buffered saline (PBS) (Fisher Scientific, Waltman, MA, USA), respectively. The amount of vehicle used to prepare the CP55,940 stock solution depended on the mouse’s body weight and designated treatment dose. TCS-1102 was first mixed in a 10% dimethylsulfoxide (DMSO) solution in PBS, and then combined with the vehicle. The final stock concentration of TCS-1102 was 1.5 mg/mL, which was further diluted in the vehicle solution accounting for mouse weight and dose. SR141716A and YNT-185 were first dissolved in 1:1:18 of 100% ethanol, Kolliphor EL, and 1 M PBS, respectively, where vehicle quantity was based on the body weight and treatment dose of the mouse. All compounds were prepared at room temperature, after which they were stored for 1–3 h at 4 °C before being injected into mice.

### 2.2. Animals

Male and female C57BL/6 mice aged 6–12 weeks (mean body weight of males: 25 ± 0.3 g; mean body weight of females: 20 ± 0.3 g) were purchased from Charles River Laboratories (Senneville, QC, Canada). Males were housed in groups of 3, while females were housed in groups of 5 with *ad libitum* access to food, water, nesting material, and environmental enrichment. All mice were maintained on a 12 h light/dark cycle (07:00–19:00/19:00–07:00) throughout the 8-day SPT and 24 h RST, which is outlined in Figure 1A. This experiment included male and female mice (*n =* 3 mice per sex) across seven treatment groups, each either receiving RST or not, for a total of 84 mice for the study (i.e., 7 treatments × 2 groups [RST vs. non-RST] × 2 sexes × *n* = 3 per sex). For every treatment, mice were intraperitoneally (i.p.) injected once on each side a total of two times: (1) before the No 24 h RST/24 h RST period, and (2) immediately prior to the final sucrose preference measurement of the SPT. CP55,940 and TCS-1102 doses were based on previous studies from our group [40,42,43,44]. The doses of SR141716A and YNT-185 were determined from past behavioural testing of these drugs in mice [42,45]. The animal care protocol abided by the guidelines of the Canadian Council on Animal Care [46] and was approved by the Animal Research Ethics Board and the Scientific Merit Review Committee for Animal Behavior at the University of Saskatchewan. In satisfying the Animal Research: Reporting of In Vivo Experiments (ARRIVE) guidelines, the minimum number of mice required for the study was determined to be *n* = 6 per group using power analysis for the SPT measurement a priori. Due to practical and personnel constraints of conducting these experiments, experimenters were not blinded to the treatment group during physiological and behavioural assessments but were blinded for immunohistochemical and molecular analyses.

### 2.3. SPT and 24 h RST

Mice were acclimatized to their home cages for 7 days, then sufficiently handled leading up to the 8-day SPT and 24 h RST. Mice were also weighed between 09:00–11:00 on each day except for days 2, 4, and 7 as this would have interfered with experimentation throughout the 6-day timeline. Before 17:00 on day 1, all SPT bottles (Flemington, NJ, USA, Cat #9019, Bio-Serv) were cleaned and sanitized with 70% ethanol and dishwashing detergent. Once dried, half of the bottles were filled with 50 mL of double-distilled water and the other half with 50 mL of 1% (wt/vol) sucrose solution. At 17:00, two bottles—one filled with water and one with sucrose solution—were placed in each home cage for a continuous 48 h, allowing for the five mice occupying each cage to acclimatize to the bottles (Figure 1A). In preparation for the next step, the SPT chambers were cleaned, sanitized, and outfitted with a sheet of paper towel and two food pellets. The bottles were taken out of the home cages, cleaned, sanitized, and refilled with either 50 mL of water or 50 mL of sucrose solution. At 17:00 on day 3, each mouse was placed in their own outfitted SPT chamber containing the two refilled bottles (Figure 1A). Mice remained in the chambers for 24 h to acclimate, during which they were monitored every 6 h for breathing and tactile response. At 17:00 on day 4, all mice were returned to their home cages (Figure 1A). Ahead of the next step, all SPT chambers and bottles were cleaned, sanitized, refilled with their respective solutions, and weighed. The first baseline sucrose preference measurement commenced at 21:00 on day 4, as each mouse was placed in its own SPT chamber containing a paper towel and the refilled bottles for 12 continuous hours (Figure 1A). At 09:00 on day 5, the bottles were weighed, and all mice were returned to their home cages (Figure 1A). Again, the chambers and bottles were cleaned, sanitized, re-prepared, and weighed. The second baseline measurement resumed at 09:00 on day 5 and lasted 12 h until 09:00 on day 6, at which point, each bottle was weighed, and all mice were returned to their home cages (Figure 1A). During these first and second 12 h baseline measurements, mice were monitored once at the halfway 6 h mark.

At 20:00 on day 6, all mice were injected with their first of 2 treatments before a subset of these mice entered the 24 h RST (Figure 1A,B). The aim of this first treatment was to pharmacologically interfere with and prevent the stress response. Each mouse also received a 200 µL subcutaneous injection of Lactated Ringer’s solution for hydration ahead of the 24 h RST. Mice were restrained using a DecapiCone (Braintree, Chicago, IL, USA, Cat # NC9679094) (Figure 1B) [13], and then each was placed in a clean, sanitized, and empty SPT chamber for the next 24 h (Figure 1A). Figure 1B illustrates the 24 h RST apparatus and protocol, which was adapted from that of Chu et al. (2016) [12]. During this prolonged restraint period, mice were monitored every 6 h. At 09:00 on day 8, mice were released from the restraint, injected with their second treatment, and then placed in the SPT containing a paper towel and the refilled, weighed bottles (Figure 1A). The aim of this second treatment was to maintain drug levels in the animal’s system for the entire duration of the test. The final preference measurement then occurred over 12 h (Figure 1A). At 09:00 on day 8, all bottles were weighed, and mice were removed from their chambers (Figure 1). Mice were euthanized before 12:00 that same day (Figure 1A). This 8-day experimental timeline was designed to resemble that of (Figure 1A) [47].

### 2.4. Tissue Perfusion and Immunohistochemistry

Following the final preference measurement, a subset of mice was anesthetized, euthanized, perfused, and their brain tissue was collected by 12:00 on the 8th experimental day. Mice were first placed in a rodent induction chamber, which administered an oxygen and isoflurane mixture for approximately 2 min before the mouse was fully anesthetized. Mice then underwent transcardial perfusions with 5 mL of ice-cold 0.9% saline solution followed by 5 mL of ice-cold 4% paraformaldehyde solution. Brains were dissected and stored in 4% paraformaldehyde solution at 4 °C for 1 day before being re-submerged in 30% sucrose solution at 4 °C for an additional 2 days (until brains sunk to the bottom of the solution). Brains were then drained, flash-frozen in liquid nitrogen, and kept at −80 °C until slicing. For cryo-slicing, the frozen brains were embedded in Tissue-Plus^TM^ O.C.T. Compound (Fisher Scientific, Waltham, MA, USA) in an appropriately sized plastic mold. A cryostat held at −20 °C was used to slice the frozen brains at a thickness of 20 µm. Slices were mounted at room temperature on Superfrost^TM^ Plus microscope slides (Fisher Scientific, Waltham, MA, USA) and then kept at −20 °C until immunohistochemistry.

Immunohistochemistry on the brain slices entailed the following steps: (1) Incubating the slices in 0.3% H_2_O_2_ at room temperature for 10 min to block endogenous peroxidase activity; (2) Rinsing in 1 M PBS 3 times for 5 min each time; (3) Incubating in 10% fetal bovine serum at room temperature for 2 h to block non-specific binding; (4) Incubating with solutions of primary antibodies at 4 °C for 24 h; (5) Rinsing again in 1 M PBS three times for 5 min each time; (6) Incubating in secondary antibody solution at room temperature for 1 h in a light-protected environment (all subsequent steps were also light-protected); (7) Rinsing once again in 1 M PBS three times for 5 min each time; (8) Coverslip mounting using ProLong^TM^ Gold Antifade Mountant with DAPI (ThermoFisher Scientific, Waltham, MA, USA). The primary antibodies are as follows: Cannabinoid Receptor CB1R Monoclonal Antibody (mouse, Synaptic Systems, Göttingen, Germany, Cat #258011, Lot 1–5) diluted in CytoVista™ Antibody Dilution Buffer (Invitrogen, Eugene, OR, Cat #V11305) at 1:500, Orexin Receptor 1 Polyclonal Antibody (rabbit, Bioss Antibodies, Woburn, MS, USA, Cat #bs-18029R, Lot BB05185553) diluted with antibody dilution buffer at 1:50, and the Orexin Receptor 2 Polyclonal (rabbit, Enzo Life Sciences, Farmingdale, NY, USA, Cat #BML-SA647-0200, Lot 12032006) diluted with antibody dilution buffer at 1:200. The following secondary antibodies were used: Goat Anti-Mouse IgG H&L (AlexaFlour^TM^, Invitrogen, Eugene, OR, USA, Cat #A11005, Lot 2352968) diluted with antibody dilution buffer at 1:500 was used for the CB1R primary antibody, while Goat Anti-Rabbit IgG H&L (AlexaFlour^TM^, Invitrogen, Eugene, OR, USA, Cat #A11008, Lot 2490730) diluted with antibody dilution buffer at 1:500 was used for the OX1R and OX2R antibodies. All brain slices were triple labeled with (1) DAPI/CB1R/OX1R or (2) DAPI/CB1R/OX2R. VECTASHIELD^®^ Antifade Mounting Medium with DAPI was purchased from Vector Laboratories (Burlingame, CA, USA, Cat #H-1200, Lot ZH0823). All immune-labelled tissue was stored at 4 °C before confocal microscopy.

### 2.5. Confocal Microscopy and Colocalization Analysis

The stained brain slices were imaged using a Zeiss LSM880 confocal microscope (Carl Zeiss, Oberkochen, Germany) equipped with Zeiss ZEN Black (version 2.3 SP1) software (Carl Zeiss, Oberkochen, Germany). At a 63x water immersion, this imaging system yielded 10 fluorescent Z-stacks encompassing an average tissue depth of 15 µm. The images were then merged and analyzed for receptor colocalization using the FIJI software package associated with ImageJ (version 2.1.0) (NIH, Bethesda, MD, USA). This software allowed for the random selection of DAPI-labelled cell nuclei. For an individual DAPI-labelled cell, Pearson’s correlation coefficients were calculated as the overlapping fluorescent signals of either CB1R and OX1R, or CB1R and OX2R in cell bodies of DAPI-stained nuclei in the VTA, MPFC, and NAC [48]. The coefficients of *n* = 10 cells in a given brain region of a single animal representing a distinct treatment group were averaged for statistical processing. Correlation coefficients represent the colocalization of labelled receptors in close proximity to one another but are not direct evidence of receptor heterodimerization.

### 2.6. Blood Collection and Corticosterone Quantification

After anesthetization and immediately prior to decapitation, a cardiac puncture was performed to collect atrial blood from each mouse. Blood was deposited in BD Vacutainer™ Plastic Blood Collection Tubes with Lithium Heparin: Hemogard (Becton Dickinson, Franklin Lakes, NJ, USA) and placed on ice before centrifugation at 2000× *g* for 10 min at 4 °C. The supernatant or plasma from each blood sample was saved and stored at −80 °C leading up to ELISA analysis. Corticosterone Competitive ELISA Kits (Invitrogen, Eugene, OR, USA, Cat #EIACORT, Lot 22CS018B) were used to quantify plasma concentrations of corticosterone from all samples. First, 50 µL of the plasma sample was added to each well in the kit-provided 96-well plates. Next, 75 µL of the 1X Assay Buffer was added, followed by 25 µL of the Corticosterone Conjugate and 25 µL of the Corticosterone Antibody. The prepared plates were tapped gently to mix, then sealed and incubated for 1 h at room temperature with shaking. After incubating, 100 µL of TMB Substrate was added to each well, following which the plates were covered and incubated for 30 min at room temperature without shaking. Finally, 50 uL of Stop Solution was added, and the plates were tapped to mix. A Cytation5 plate reader (BioTek, Winooski, VT, USA) programmed at an absorbance level of 450 nm was used to measure well absorbance. The blood samples of *n* = 6 mice (*n* = 3 males, *n* = 3 females) were analyzed for treatment groups: Vehicle, 3 mg/kg SR141716A, 40 mg/kg YNT-185, and 3 mg/kg SR141716A + 40 mg/kg YNT-185. The samples of *n* = 4 mice (*n* = 2 males, *n* = 2 females) were measured for groups: 0.3 mg/kg CP55,940, 1 mg/kg TCS-1102, 0.3 mg/kg CP55,940 + 1 mg/kg TCS-1102. This variance in group sizes was due to limited materials, specifically sample space in the 96-well plates. The treatment groups that were processed in this assay were prioritized based on their behavioural results.

### 2.7. Statistical Analysis

SPT data are presented as a mean ± SEM where *n* represents the number of animals in each experimental group. The four SPT datasets include (1) sucrose preference (%), (2) raw sucrose solution consumption during the final preference measurement (mg), (3) raw water consumption during the final preference measurement (mg), and (4) raw total liquid consumption during the final preference measurement (mg). Sucrose preference was calculated by the formula: [sucrose consumption in the final measurement/(sucrose consumption in the final measurement + water consumption in the final measurement)] × 100% [47]. Body weights were recorded as g and graphed as a daily change in weight since day 1. The plasma corticosterone ELISA results are displayed as absorbances in pg/mL. The SPT, body weight, and plasma corticosterone data are calculated as mean ± SEM, where *n* represents the number of animals in each experimental group. The immunohistochemistry and colocalization results are displayed as a mean ± SEM where *n* represents individual cells identified in a specific brain region of a single mouse. Colocalization means are denoted as Pearson’s correlation coefficients. Determination of statistical significance for all results, except body weights, was done using a two-way analysis of variance (ANOVA) (treatment x restraint) followed by a Tukey’s post-hoc analysis accounting for No 24 h RST/24 h RST and treatment group. Significance was set a *p* < 0.05. Determination of statistical significance for body weights was done using a repeated measures two-way ANOVA followed by a Tukey’s post-hoc analysis with repeated measurements within groups between days and across groups within days. Significance was set a *p* < 0.05. All ANOVA and post-hoc statistics can be found in Appendix A.

## 3. Results

### 3.1. Anhedonia

The SPT was used to measure changes in hedonic sucrose consumption or preference in male and female mice subjected to 24 h RST and various drug treatments (Figure 2). The 24 h RST reduced sucrose preference in mice injected with vehicle, 0.3 mg/kg CP55,940 (cannabinoid receptor agonist), 1 mg/kg TCS-1102 (DORA), 0.3 mg/kg CP55,940 + 1 mg/kg TCS-1102, and 3 mg/kg SR141716A (CB1R antagonist) compared to non-restrained mice (*p* < 0.001) (Figure 2A). Conversely, treatment with 40 mg/kg YNT-185 (OX2R agonist) and the drug combination of 3 mg/kg SR141716A + 40 mg/kg YNT-185 improved sucrose preference in mice that underwent the 24 h RST (Figure 2A). There were no treatment-dependent differences among the non-restraint mice (*p* > 0.05) (Figure 2A).

Within the 24 h RST groups, mice treated with 3 mg/kg SR141716A, 40 mg/kg YNT-185, and the combination of 3 mg/kg SR141716A + 40 mg/kg YNT-185 displayed greater sucrose preference as compared to those that received vehicle (*p* < 0.001) (Figure 2A). No sex-dependent differences were observed within non-RST nor RST groups for sucrose preference (Figure 2A). These data indicate that activating OX2R alone or in combination with CB1R blockade prevented 24 h RST-induced anhedonia in male and female mice (Figure 2A). Additionally, OX2R agonists, CB1R antagonists, or co-administration of these drugs were associated with reduced anhedonia as compared to vehicle treatments under stressful conditions (Figure 2A).

The sucrose preference data was parsed into raw sucrose, water, and total liquid consumption to visualize hedonic feeding behaviour (Figure 2B–D). Congruent with the data presented in Figure 2A, the prolonged restraint stress decreased sucrose consumption within the vehicle (*p* < 0.001), 0.3 mg/kg CP55,940 (*p* < 0.01), and 3 mg/kg SR141716A groups (*p* < 0.001) treatment groups (Figure 2B). Mice that received 1 mg/kg TCS-1102, 0.3 mg/kg CP55,940 + 1 mg/kg TCS-1102, and 40 mg/kg YNT-185 displayed a similar trend; however, these differences were not statistically significant (*p* > 0.05) (Figure 2B). With regards to raw water consumption, there were no significant differences between non-24 h RST and 24 h RST groups, nor were there differences between the drug treatments within non-restraint stressed or prolonged restraint stress groups (*p* > 0.05) (Figure 2C). Again, 24 h RST reduced the sum of sucrose and water (total liquid) consumption in the vehicle (*p* < 0.001), 0.3 mg/kg CP55,940 (*p* < 0.01), and 3 mg/kg SR141716A-treated mice (*p* < 0.001) (Figure 2D). Moreover, non-restraint mice injected with 3 mg/kg SR141716A exhibited higher liquid consumption compared to their combination drug-treated counterparts (*p* < 0.01) (Figure 2D). The lack of group differences in Figure 2C supports the notion that total liquid consumption is driven by sucrose preference over water (Figure 2).

### 3.2. Body Weight

Body weights were recorded on days 1, 3, 5, 6 and 8 to better understand the relationships between endocannabinoid- and orexin-mediated stress; hedonic feeding; and the metabolic-excretory consequences of sucrose consumption under prolonged restraint stress. Mice in all groups exhibited a loss of weight during the acclimatization period (days 3–5) that was not different between treatments or groups (Figure 3A,B). Drug treatment-dependent differences were not observed within both non-restraint and prolonged restraint stress groups on the same experimental day (Figure 3A,B). Furthermore, there were no differences between sexes nor drug treatments within non-RST and RST groups, nor were there differences between restraint and non-restraint groups within drug treatments by day 8 when mice were subjected to stress intervention (Figure 3C). Following day 8, mice subjected to RST may have experienced weight loss compared to their non-RST counterparts; however, this was not assessed because mice were euthanized for immunohistochemical analyses. Changes in weight following prolonged restraint and cannabinoid and orexin drug treatments should be explored in the future.

### 3.3. Plasma Corticosterone

Circulating corticosterone is indicative of physiological and psychological stress-HPA axis activation [49]. Although a trend between non-restraint and restraint mice was observed where the samples from the latter appeared to have higher corticosterone levels, there were no statistically significant differences between the sexes or any of the other experimental groups (Figure 4).

### 3.4. Cannabinoid–Orexin Receptor Colocalization in Male Brain Regions

#### 3.4.1. CB1R-OX1R Colocalization

Receptor colocalization measures the physical closeness or proximity of receptor proteins in a two-dimensional frame. Colocalization may therefore be indicative of potential interactions between receptors but does not provide direct evidence of heterodimerization [50]. In the current study, prolonged restraint stress along with drug treatments altered cannabinoid and orexin receptor colocalization levels within mesocorticolimbic regions involved in hedonic feeding [51]. Based on the SPT results (Figure 2), CB1R-OX1R/OX2R colocalization was quantified in the VTA, NAC, and MPFC of the vehicle, 3 mg/kg SR141716A, 40 mg/kg YNT-185, and 3 mg/kg SR141716A + 40 mg/kg YNT-185-treated male and female mice (Figure 5, Figure 6, Figure 7 and Figure 8). In the VTA of male mice, 3 mg/kg SR141716A + 40 mg/kg YNT-185 led to greater CB1R-OX1R colocalization as compared to 3 mg/kg SR141716A and vehicle treatments in non-restraint samples (*p* < 0.01) (Figure 5A). Within the 24 h RST groups, the combination of 3 mg/kg SR141716A + 40 mg/kg YNT-185 resulted in the most CB1R-OX1R colocalization amongst all other drug treatments (*p* < 0.01–0.001) (Figure 5A). Similarly, in the male NAC, 3 mg/kg SR141716A + 40 mg/kg YNT-185 increased CB1R-OX1R colocalization relative to 40 mg/kg YNT-185, 3 mg/kg SR141716A, and vehicle-treated samples amongst the non-24 h RST groups (*p* < 0.001) (Figure 5B). Within the 24 h RST groups, 3 mg/kg SR141716A + 40 mg/kg YNT-185 led to greater CB1R-OX1R colocalization compared to samples treated with 40 mg/kg YNT-185 (*p* < 0.05) and 3 mg/kg SR141716A (*p* < 0.001) (Figure 5B). The MPFC of males showed a different trend wherein 3 mg/kg SR141716A + 40 mg/kg YNT-185 resulted in the highest degree of CB1R-OX1R colocalization in only the non-restraint samples (*p* < 0.001) (Figure 5C). Amongst the 24 h RST groups, vehicle-induced CB1R-OX1R colocalization was larger than that of 3 mg/kg SR141716A (*p* < 0.001) (Figure 5C). Also, within the vehicle group, 24 h RST increased CB1R-OX1R colocalization (*p* < 0.05) (Figure 5C). Representative images of CB1R-OX1R colocalization in the male VTA, NAC, and MPFC of 24 h RST groups are presented in Figure 5D–F; representative images from No-RST groups are presented in Appendix A. These colocalization results serve as evidence for receptors existing near each other within the same cells but do not prove that the receptor proteins are physically nor functionally interacting on the plasma membrane.

#### 3.4.2. CB1R-OX2R Colocalization

Colocalization analysis was also done for CB1R and OX2R. In the VTA of males subjected to 24 h RST, the combination drug treatment of 3 mg/kg SR141716A + 40 mg/kg YNT-185 increased CB1R-OX2R colocalization as compared to 3 mg/kg SR141716A (*p* < 0.05) (Figure 6A). In the NAC of non-restraint groups, the CB1R-OX2R colocalization of vehicle samples was higher than that of 3 mg/kg SR141716A + 40 mg/kg YNT-185 (Figure 6B). Prolonged restraint stress also increased CB1R-OX2R colocalization in the male NAC of 40 mg/kg YNT-185-treated samples (*p* < 0.05) (Figure 6B). No differences were detected between experimental groups in the MPFC (*p* > 0.05) (Figure 6C). Representative images of CB1R-OX2R colocalization throughout these mesocorticolimbic brain regions are shown in Figure 6D–F; representative images from No-RST groups are presented in Appendix A.

### 3.5. Cannabinoid–Orexin Receptor Colocalization in Female Brain Regions

#### 3.5.1. CB1R-OX1R Colocalization

Cannabinoid and orexin receptor colocalization was measured in female brain samples. In the female VTA of samples subjected to 24 h RST, 3 mg/kg SR141716A + 40 mg/kg YNT-185 led to the greatest level of CB1R-OX1R colocalization among all other drug treatment groups (*p* < 0.001) (Figure 7A). Additionally, within this combination group, 24 h RST increased CB1R-OX1R colocalization (*p* < 0.05), while no other drug treatments caused such a pattern (Figure 7A). The female NAC did not depict any statistically significant differences between experimental groups (*p* < 0.05) (Figure 7B). In the MPFC, 24 h RST elevated CB1R-OX1R colocalization in the 3 mg/kg SR141716A + 40 mg/kg YNT-185 group (*p* < 0.001) (Figure 7C). The female MPFC saw no other differences between experimental groups (*p* > 0.05) (Figure 7C). Figure 7D–F offers representative images of CB1R-OX1R colocalization throughout the VTA, NAC, and MPFC of these experimental groups; representative images from No-RST groups are presented in Appendix A.

#### 3.5.2. CB1R-OX2R Colocalization

In terms of CB1R-OX2R colocalization in females, there were no differences between non-restraint-stressed and prolonged restraint stress nor between drug treatment groups (*p* > 0.05) (Figure 8A–C). Figure 8D–F offers representative images of CB1R-OX2R colocalization throughout the VTA, NAC, and MPFC of these experimental groups; representative images from No-RST groups are presented in Appendix A.

## 4. Discussion

To evaluate cannabinoid and orexin receptor-mediated anhedonic and hedonic responses in the current study, a prolonged restraint stress model was used to decrease sucrose preference in male and female C57BL/6 mice. The 24 h RST served as both (1) a deprivation period leading up to the final sucrose preference measurement and (2) a severe enough stressor that would guarantee an anhedonic phenotype in mice. This model was adapted from Chu et al. (2016), who revealed short- and long-term decreases in sucrose preference as well as learned helplessness in mice subjected to the 24 h RST [12]. Along with these behavioural changes, serum corticosterone increases during and immediately after the 24 h restraint period [12]. Throughout the next 24 h following release from the RST, serum corticosterone levels rapidly return to normal in the study by Chu and colleagues (2016) [12]. Similar observations were made in the current study, in which plasma corticosterone was not significantly different between non-restraint and restraint groups by the morning of day 8 (12 h following release from the RST) (Figure 4). In terms of body weight, Chu and colleagues (2016) reported weight loss immediately following the 24 h RST, which was also restored to normal by the next day [12]. Weight loss did not differ between any of the experimental groups in the current study (Figure 3). Although this restraint stress test was intensely stressful, mice were still able to partake and be behaviourally evaluated in the SPT.

The current dataset is the first to assess dual cannabinoid and orexin receptor functionality following prolonged restraint stress. The ECS is involved in the stress response as it buffers HPA axis over-activation to prevent maladaptive behaviours [51]. Although stress-induced endocannabinoid activity is brain-region-specific, AEA levels generally exhibit an immediate decrease, while 2-AG levels increase over time or after the stressful stimuli to reinstate parasympathetic tone [51,52,53,54,55]. Chronic stress often leads to anhedonia, which has been shown to be worsened by concomitant genetic or pharmacological disruption of CB1R signaling [56,57]. Interestingly, CB1R blockade by SR141716A partially improved 24 h RST-induced anhedonia in the current study relative to vehicle treatment but not non-restrained SR141716A-treated mice (Figure 2). This result raises questions about how the 24 h RST differs from other chronic stress paradigms. Specifically, 24 h of food, water, and movement restriction elicited unique changes in the ECS, after which SR141716A rescued hedonic feeding. While anhedonia was the only behaviour evaluated, it remains unknown whether CB1R blockade may have also improved other depression-like and reward-related behaviours following the 24 h RST.

Unlike the ECS, physiological and psychological stress recruit orexin neurons to stimulate the release of the corticotropin-releasing hormone to initiate HPA axis activation [58]. Acute stressors (e.g., i.p. injection, exercise, forced swim test, 30-min restraint) elevate orexin levels and neuronal activation to encourage behavioural adaptation, active coping via hedonic feeding, and resilience [59,60,61]. Hyperactivation of the orexin system can, however, create anxious phenotypes in animals and humans [62,63,64]. In terms of repeated or chronic stress, this can result in orexin hypoactivity, passive coping, and depressive symptoms, including low-motivated behaviours [65,66,67]. Based on this literature and the incidence of OX2R stimulation improving sucrose preference in the current study (Figure 2), 24 h of restraint stress likely decreased orexin activation and induced maladaptive stress habituation. SR141716A, as well as the combination of YNT-185 and SR141716A, improved chronic stress-induced anhedonia despite what is reported in the literature, which posits the interdependent and orexin-driven nature of these systems under such stressful conditions.

Our receptor colocalization studies provide additional insight into the physical and functional interactions between these neuromodulatory systems. As an OX2R agonist, YNT-185 treatment was associated with greater CB1R-OX1R colocalization as compared to CB1R-OX2R (Figure 5, Figure 6, Figure 7 and Figure 8). After two consecutive 40 mg/kg doses of YNT-185, persistent OX2R stimulation may have led to receptor downregulation and compensatory upregulation of OX1R to which endogenous orexins preferentially activated. Alternatively, increased CB1R-OX1R colocalization induced by the combination of SR141716A and YNT-185 may have resulted from cannabinoid receptor blockade such that SR141716A provoked the upregulation of heterodimerized CB1R-OX1R. Although brain regions were not directly compared, the MPFC was seemingly less sensitive to both prolonged restraint stress and drug interventions as compared to the VTA and NAC (Figure 5 and Figure 7). A recent circuitry investigation reported that larger proportions of orexin-innervated dopaminergic neurons project to the NAC relative to the MPFC [68]. Finally, these subtypes- and region-dependent results may be due to the relative basal abundances of these receptors. The NAC is denser in OX2R, whereas the MPFC expresses equal levels of both receptor subtypes [38,69]. Higher OX2R expression of the NAC may have driven detectable differences between drug treatments in the current study.

Notwithstanding the lack of sex-specific differences in the behavioural and physiological measures, stress-induced receptor colocalization was more evident in male mesocorticolimbic regions as compared to females. In our previous drug–drug interaction study, male mice were more sensitive to the hypolocomotive effects of CP55,940 and TCS-1102, an observation supported by greater CB1R-OX1R colocalization in their ventral striatum, which encapsulates the NAC [40]. CB1R density and binding are higher throughout the cortex and amygdala of male rats versus estrogen-dominant females [70]. In other regions, such as the caudate-putamen, NAC, and hippocampus, this sex difference is negligible [70].

In terms of the orexin system, the brains of female rodents exhibit greater mRNA expression and protein concentrations of orexins [71]. Data comparing the brain region-specific abundance and activity of orexin receptors between male and female mice have yet to be procured. It also remains unknown whether cannabinoid and orexin receptor heterodimers are subject to such region-specific sex dimorphisms. Aside from testing animals, futures studies should evaluate the downstream signaling effects of sex hormones, such as estrogen, progesterone, and testosterone, in cell cultures known to express cannabinoid and orexin heterodimers. These experiments may reveal more about how the cannabinoid and orexin receptors are impacted by the neuroendocrine system.

The current study showed that dual cannabinoid activation and orexin blockade leads to additive sedative effects [40]. Both arousal/sedation and reward processing are dependent on striatal circuitry [72,73,74], yet the opposite combination of pharmacological agents differentially mediated these functions according to past and current experiments [40,75]. Thus, subsequent research into ECS-orexin interactions should delineate how endocannabinoids fine-tune excitation and inhibition within orexin neurons in the striatum. These neurons operate in a complex network of overlapping efferent and afferent brain regions mediating the multimodal stress response, suggesting that much like the ECS, the orexin system both processes incoming stressful stimuli, and is involved in the adaptive responses to stressful experiences [67,76].

In consideration of these results, two key limitations need to be addressed. First, this study utilized a single behavioural readout for anhedonia, the SPT. This may be problematic for the reason that both cannabinoid and orexin signaling are involved in appetite reinforcement. In addition, orexin-modulating drugs may alter wakefulness, which likely affected the current results. Future investigations exploring the interactions between these systems should incorporate additional measures of anhedonia, which were not possible here due to experimental and financial limitations. Second, only one dose for each drug was studied. While all doses were selected based on past experiments testing these same drugs, subsequent studies should assess the dose-response relationship with respect to anhedonia, particularly for the CB1R antagonist SR141716A and OX2R agonist YNT-185.

## 5. Conclusions

Anhedonia is a display of natural reward dysfunction that is commonly seen in major depressive disorder, among other mental illnesses. The endocannabinoid and orexin systems are involved in the initial response and adaptation to stressful stimuli. Thus, dual pharmacological manipulation of these systems greatly impacts stress-induced anhedonia. While the 24 h RST likely downregulated orexin signaling, as past chronic stress studies have also shown, the ECS exemplified a unique tone to which the CB1R blockade markedly improved anhedonia. Treatment with both YNT-185 and SR141716A also increased CB1R-OX1R colocalization in select mesocorticolimbic brain regions. These results shed new light on the physical and functional interactions between the endocannabinoid and orexin systems.

## Figures and Tables

**Figure 1 brainsci-13-00314-f001:**
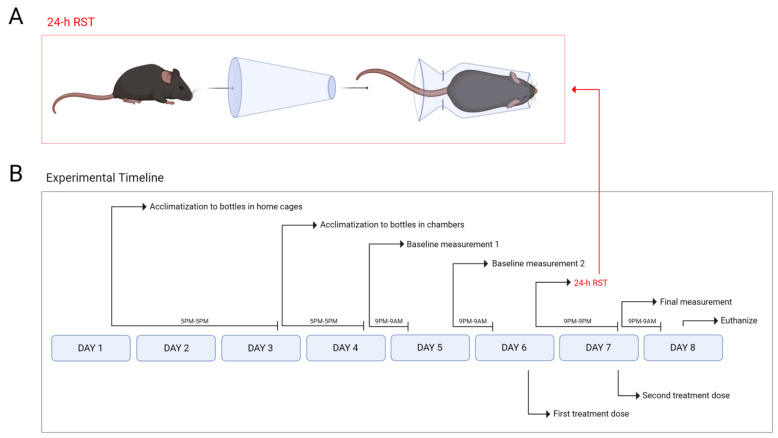
Experimental methods and timeline. (**A**) 24 h RST using a DecapiCone restraint apparatus. Mice were placed in the cone such that their noses poked out the opening. Metal staples were used to tightly secure mice within the cone. Mice could not move around; however, they were easily able to breathe. (**B**) Experimental timeline for the SPT including the 24 h RST. From 17:00 on day 1 to 17:00 on day 3, all mice were acclimatized to the water and sucrose solution-containing bottles in their home cages for 48 h. From 17:00 on day 3 to 17:00 on day 4, mice habituated to the SPT chambers containing the water and sucrose solution-filled feeders for 24 h. At 21:00 on day 4, mice underwent the first of two baseline sucrose preference measurements that lasted 12 h. At 21:00 on day 5, the second baseline measurement was carried out for 12 h. Immediately prior to the 24 h RST, all mice were injected with their designated drug treatments. From 21:00 on day 6 to 21:00 on day 7, a subset of mice were restrained in DecapiCones for 24 h. Following the 24 h RST, all mice were treated a second time before entering the final sucrose preference measurement, which lasted for 12 h. Euthanasia and tissue collection were completed by 12:00 on day 8.

**Figure 2 brainsci-13-00314-f002:**
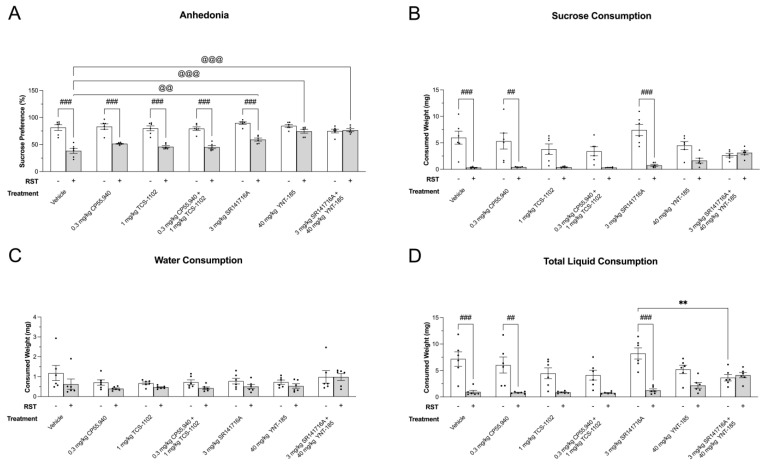
Cannabinoid and orexin drug effects on 24 h RST-induced anhedonia. (**A**) Sucrose preference of 200 µL of vehicle, 0.3 mg/kg CP55,940 (cannabinoid receptor agonist), 1 mg/kg TCS-1102 (DORA), 0.3 mg/kg CP55,940 + 1 mg/kg TCS-1102, 3 mg/kg SR141716A (CB1R antagonist), 40 mg/kg YNT-185 (OX2R agonist), and 3 mg/kg SR141716A + 40 mg/kg YNT-185-treated adult male and female C57BL/6 mice. An 8-day SPT, which included a 24 h RST was used to obtain the sucrose preference data expressed as %. (**B**) Raw sucrose solution consumption of mice is expressed as (mg). (**C**) Raw water consumption of mice expressed as (mg). (**D**) Raw total solution consumption of mice is expressed as (mg). *n* = 6 mice (*n* = 3 males; 3 females). Means ± SEM were used for statistical significance analysis via a two-way ANOVA followed by a Tukey’s post-hoc analysis. ** *p* < 0.01 compared between drug treatments within No 24 h RST. ^@@^/^@@@^ *p* < 0.01/0.001 compared between drug treatments within 24 h RST. ^##^/^###^ *p* < 0.01/0.001 compared between No 24 h RST and 24 h RST within drug treatments. In the figure above “+” refers to with RST and “−“ refers to without RST.

**Figure 3 brainsci-13-00314-f003:**
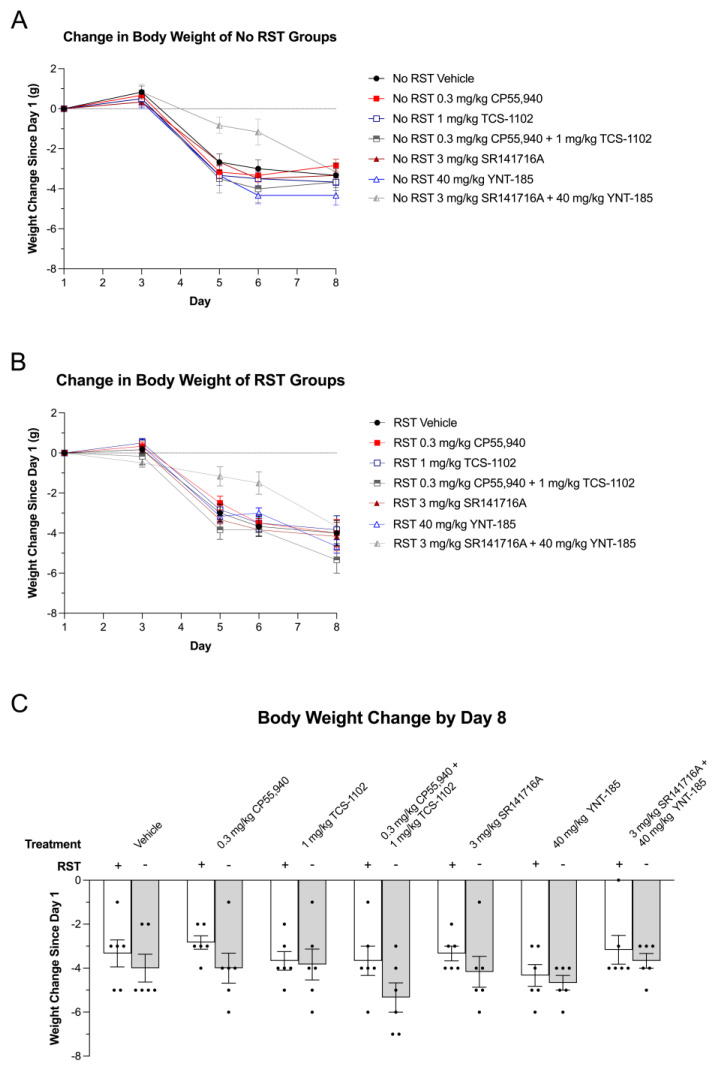
Change in body weight of cannabinoid and orexin drug-treated mice throughout the SPT and 24 h RST. (**A**) Daily change in body weight since day 1 of No 24 h RST groups. (**B**) Daily change in body weight since day 1 of RST groups. (**C**) Body weight change between days 1–8 of all drug and stress treatment groups. (**A**–**C**) The body weights of adult male and female C57BL/6 mice treated with either 200 µL of vehicle, 0.3 mg/kg CP55,940 (cannabinoid receptor agonist), 1 mg/kg TCS-1102 (DORA), 0.3 mg/kg CP55,940 + 1 mg/kg TCS-1102, 3 mg/kg SR141716A (CB1R antagonist), 40 mg/kg YNT-185 (OX2R agonist), and 3 mg/kg SR141716A + 40 mg/kg YNT-185 were measured in the morning of days 1, 3, 5, 6, and 8. Each day’s body weight was compared to that of day 1. Change in body weight is expressed as (g). *n* = 6 mice (*n* = 3 males; *n* = 3 females) as each experimental group is reported as a mean ± SEM. Statistical significance for graphs A and B was assessed by a two-way ANOVA followed by a Tukey’s post-hoc analysis with repeated measurements within groups between days and across groups within days. Significant differences within panel **C** were evaluated via Tukey’s post-hoc analysis without repeated measures. In the figure above “+” refers to with RST and “−“ refers to without RST.

**Figure 4 brainsci-13-00314-f004:**
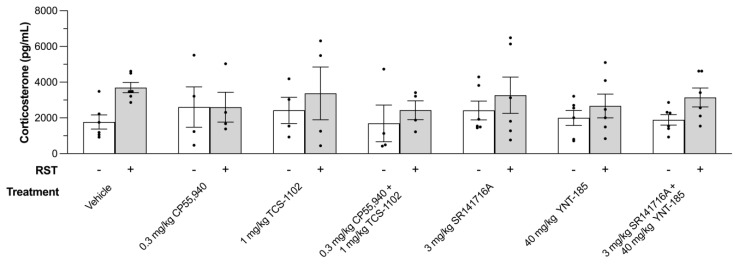
The effect of cannabinoid and orexin drugs on plasma corticosterone levels of mice that underwent the SPT and 24 h RST. After the final sucrose preference measurement on day 8, adult male and female C57BL/6 mice were euthanized, and their blood was collected for quantification of the circulating corticosterone using an ELISA kit. Data from 200 µL vehicle, 0.3 mg/kg CP55,940 (cannabinoid receptor agonist), 1 mg/kg TCS-1102 (DORA), 0.3 mg/kg CP55, 940 + 1 mg/kg TCS-1102, 3 mg/kg SR141716A (CB1R antagonist), 40 mg/kg YNT-185 (OX2R agonist), and 3 mg/kg SR141716A + 40 mg/kg YNT-185-treated mice are expressed as pg/mL. *n* = 6 mice (*n* = 3 males, *n* = 3 females) for Vehicle, 3 mg/kg SR141716A, 40 mg/kg YNT-185, and 3 mg/kg SR141716A + 40 mg/kg YNT-185 treatment groups. *n* = 4 mice (*n* = 2 males, *n* = 2 females) for 0.3 mg/kg CP55,940, 1 mg/kg TCS-1102, 0.3 mg/kg CP55,940 + 1 mg/kg TCS-1102 treatment groups. Means ± SEMs were used to evaluate statistical significance by a two-way ANOVA followed by a Tukey’s post-hoc analysis. In the figure above “+” refers to with RST and “−“ refers to without RST.

**Figure 5 brainsci-13-00314-f005:**
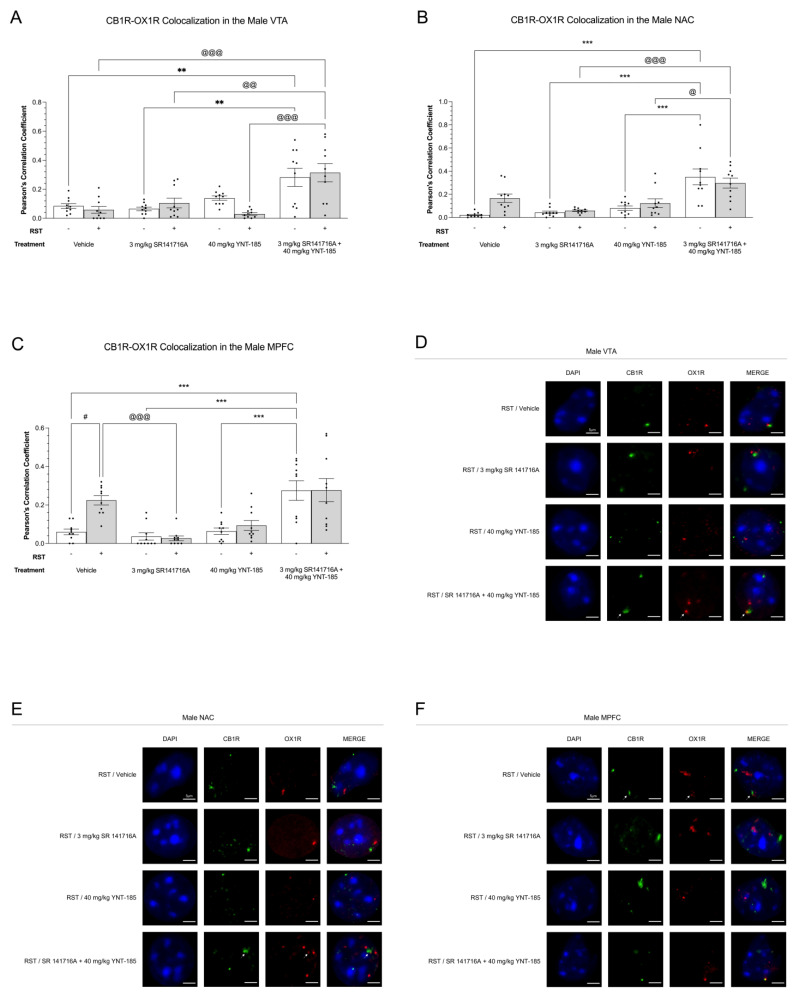
Colocalization between CB1R and OX1R in male mesocorticolimbic brain regions following prolonged restraint stress and drug treatments. (**A**–**C**), CB1R-OX1R colocalization in the VTA (**A**,**D**), NAC (**B**,**E**), and MPFC (**C**,**F**) of 200 µL vehicle, 3 mg/kg SR141716A, 40 mg/kg YNT-185, and 3 mg/kg SR141716A + 40 mg/kg YNT-185-treated adult male C57BL/6 mice. Immediately following the final sucrose preference measurement on day 8, mice were euthanized, and their brains were perfused for immunohistochemical analysis. A Zeiss LSM880 confocal microscope was used to collect the colocalization images. FIJI with ImageJ (version 2.1.0) was used to analyze the images. Receptor colocalization data are expressed as Pearson’s correlation coefficients for *n* = 10 cells per treatment group randomly selected from three male mice. Means ± SEM were used to calculate statistical significance via a two-way ANOVA and Tukey’s post-hoc analysis. **/*** *p* < 0.01/0.001 compared between drug treatments within No 24 h RST. ^@^/^@@^/^@@@^ *p* < 0.05/0.01/0.001 compared between drug treatments within 24 h RST. # *p* < 0.05 compared between No 24 h RST and 24 h RST within drug treatments. (**D**–**F**), Representative confocal microscopy images of CB1R-OX1R colocalization in the VTA (**D**), NAC (**E**), and MPFC (**F**) of males subjected to RST, which correspond to graphs A, B, and C, respectively. In the figure above “+” refers to with RST and “‒“ refers to without RST.

**Figure 6 brainsci-13-00314-f006:**
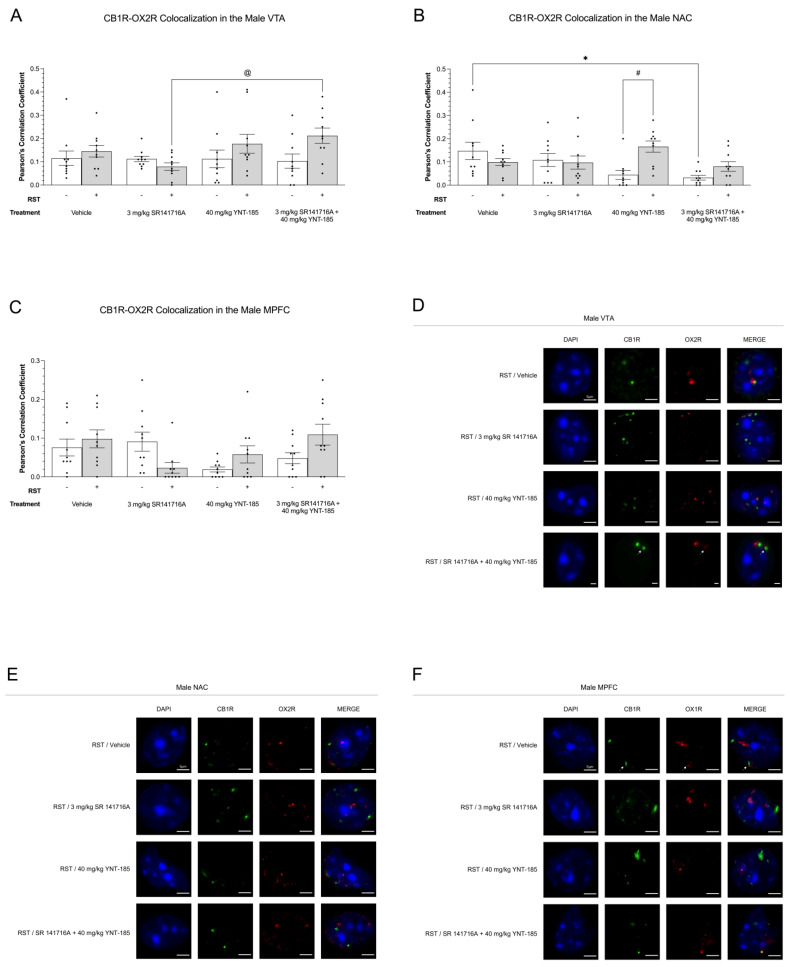
Cannabinoid receptor type 1 and OX2R colocalization in mesocorticolimbic brain regions of male mice that had undergone 24 h RST and drug treatments. (**A**–**C**), CB1R-OX2R colocalization in the VTA (**A**,**D**), NAC (**B**,**E**), and MPFC (**C**,**F**) of 200 µL vehicle, 3 mg/kg SR141716A, 40 mg/kg YNT-185, and 3 mg/kg SR141716A + 40 mg/kg YNT-185-treated adult male C57BL/6 mice. Succeeding the final sucrose preference measurement on day 8, mice were euthanized for their brains to be perfused for immunohistochemistry. A Zeiss LSM880 confocal microscope obtained the receptor colocalization images. Analysis of these images was done via FIJI and ImageJ (version 2.1.0). Colocalization data are expressed as Pearson’s correlation coefficients for *n* = 10 cells per treatment group randomly selected from three male mice. Means ± SEM were used to determine statistical significance using a two-way ANOVA and Tukey’s post-hoc analysis. * *p* < 0.05 compared between drug treatments within No 24 h RST. @ *p* < 0.05 compared between drug treatments within 24 h RST. # *p* < 0.05 compared between No 24 h RST and 24 h RST within drug treatments. (**D**–**F**), Representative images of CB1R-OX2R colocalization in the VTA (**D**), NAC (**E**), and MPFC (**F**) of males subjected to RST, which associate with graphs (**A**–**C**), respectively. In the figure above “+” refers to with RST and “‒“ refers to without RST.

**Figure 7 brainsci-13-00314-f007:**
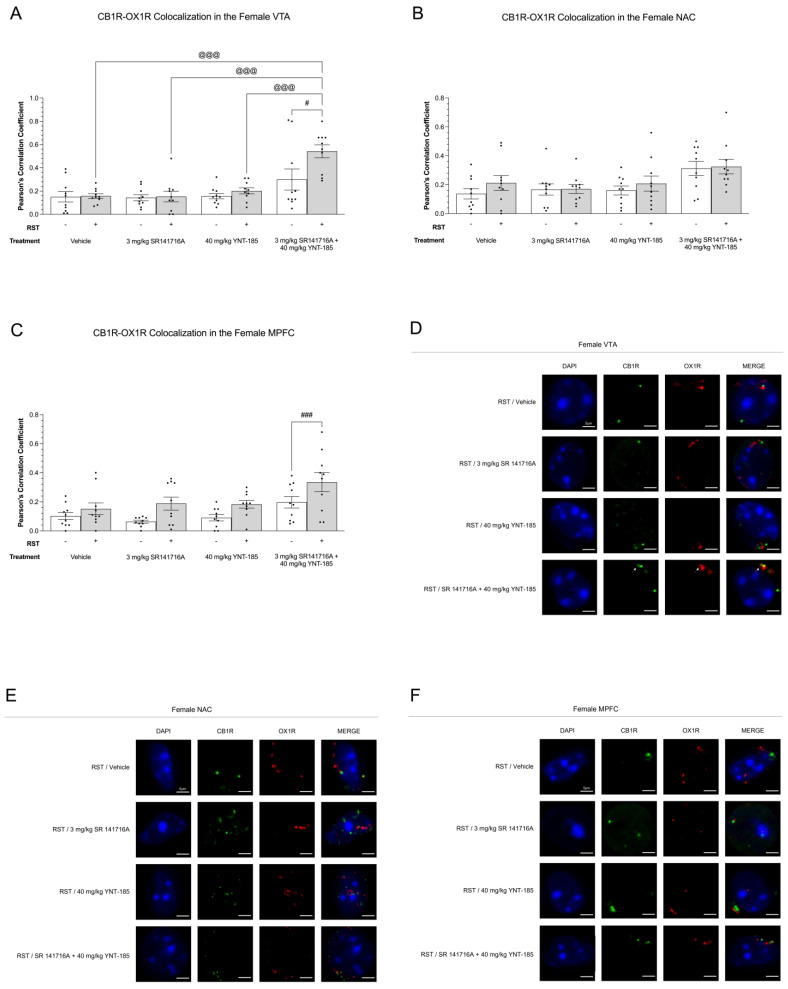
Colocalization of CB1R and OX1R in female mesocorticolimbic brain regions as a result of prolonged restraint stress and drug treatments. (**A**–**C**), CB1R-OX1R colocalization in the VTA (**A**,**D**), NAC (**B**,**E**), and MPFC (**C**,**F**) of 200 µL vehicle, 3 mg/kg SR141716A, 40 mg/kg YNT-185, and 3 mg/kg SR141716A + 40 mg/kg YNT-185-treated adult female C57BL/6 mice. On day 8, following the final sucrose preference measurement, mice were anesthetized and perfused in order to collect brain tissue for immunohistochemical analysis. Colocalization images were taken using a Zeiss LSM880 confocal microscope. Images were then analyzed on the FIJI extension of ImageJ (version 2.1.0). Colocalization data are expressed as Pearson’s correlation coefficients for *n* = 10 cells per treatment group randomly selected from three female mice. Means ± SEM were calculated, and a two-way ANOVA followed by a Tukey’s post-hoc analysis was used to determine statistical significance. ^@@@^ *p* < 0.001 compared between drug treatments within 24 h RST. ^#^/^###^ *p* < 0.05/0.001 compared between No 24 h RST and 24 h RST within drug treatments. (**D**–**F**), Representative confocal microscopy images of CB1R-OX1R colocalization in the VTA (**D**), NAC (**E**), and MPFC (**F**) of females subjected to RST, which correspond to graphs (**A**–**C**), respectively. In the figure above “+” refers to with RST and “‒“ refers to without RST.

**Figure 8 brainsci-13-00314-f008:**
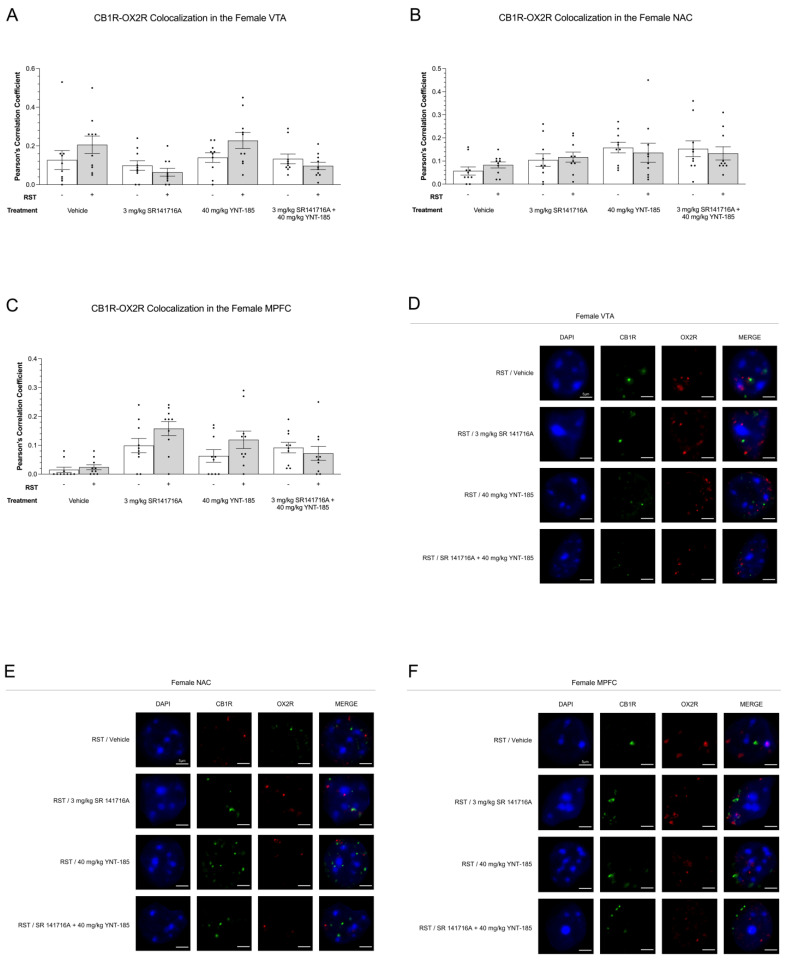
Cannabinoid receptor type 1 and OX2R colocalization throughout mesocorticolimbic brain regions of female mice subjected to 24 h RST and cannabinoid and orexin drug treatments. (**A**–**C**), CB1R-OX2R colocalization in the VTA (**A**,**D**), NAC (**B**,**E**), and MPFC (**C**,**F**) of 200 µL vehicle, 3 mg/kg SR141716A, 40 mg/kg YNT-185, and 3 mg/kg SR141716A + 40 mg/kg YNT-185-treated adult female C57BL/6 mice. Mice were euthanized, and their perfused brains were collected following the final sucrose preference measurement on day 8. Immunohistochemical analysis was performed on perfused brain slices, which were then imaged using a Zeiss LSM880 confocal microscope. Receptor colocalization analysis was done using FIJI and ImageJ (version 2.1.0). Data are expressed as Pearson’s correlation coefficients for *n* = 10 cells per treatment group randomly selected from three female mice. Means ± SEM allowed for statistical significance analysis using a two-way ANOVA and Tukey’s post-hoc analysis. (**D**–**F**), Representative images of CB1R-OX2R colocalization in the VTA (**D**), NAC (**E**), and MPFC (**F**) of females subjected to RST, which associate with graphs (**A**–**C**), respectively. In the figure above “+” refers to with RST and “‒“ refers to without RST.

## Data Availability

The data presented in this study are available on request from the corresponding author. All statistical analyses of data are provided in Appendix A.

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
