# Peer review of "Dual Cannabinoid and Orexin Regulation of Anhedonic Behaviour Caused by Prolonged Restraint Stress"

_brainsci, 2023, doi:10.3390/brainsci13020314_

Round 1
Reviewer 1 Report
This manuscript aimed to examine the cannabinoid-orexin receptor function in 24-h restrain stress-induced anhedonic behavior in male and female mice. The authors used sucrose preference test to access the anhedonic behavior. They first collected the baseline sucrose preference data. They then administered various cannabinoid and orexin receptor agonist or antagonists or combination of these agonist and antagonist to the mice right before the 24-h restrain stress. After the restrain stress, the mice were given the second drug injection then enter the final sucrose preference test. Afterwards the mice were euthanized and the blood and brain samples were collected. Blood samples were used to quantify plasma corticosterone levels. Brain samples were used in immunohistochemical staining experiments to localize the cannabinoid receptor type 1 and orexin receptor type 1 or orexin receptor type 2. The results indicated that YNT-185, the OXR2 agonist and the drug combination of CB1R antagonist SR141716A and TSC-1102 (here is a mistake in the text on page 6, line 283; it should be YNT-185) improved sucrose preference in stressed mice. There is no significant difference in body weight and plasma corticosterone levels in different treatment groups. Cannabinoid-orexin receptor co-localization study in male mice showed that there was increased CB1R and OXR1 co-localization in VTA and NAc of SR131716A+YNT-185 treated groups. In the mPFC, restrain stress increased the CB1R and OXR1 co-localization in vehicle treated group, and only the non-restraint mice showed increased co-localization by SR131716A+YNT-185 treatment. As for the co-localization of CB1R and OXR2, the SR132716A + YNT-185 treatment group showed significantly more co-localization than the SR141716A treatment group in the VAT. In the NAc, SR131716A+YNT-185 treatment reduced the CB1R and OXR2 co-localization in the non-restrained group. Restraint stress increased CB1R and OXR2 co-localization in YNT-185 treated group compared to non-restrain YNT-185 treated mice (However, the text said reduce, Page 12, line 421). There is no difference in mPFC. In the female mice, SR131716A+YNT-185 treatment significantly increased CB1R and OXR1 co-localization in the VTA. The authors wrote “ Additionally, within this combination group, 24-h RST decreased CB1R-OX1R colocalization (p< 0.05) while no other drug teratments caused such a pattern (Figure 7A).” on page 12, line 432. However, according to Figure 7A, the restraint group showed increased co-localization, not decreased! Also, the authors wrote that “In the MPFC, 24-h RST lessened CB1R-OX1R colocalizaiton in the 3mg/kg SR131716A + 40 mg/kg YNT-185 group (p< 0.001)(Figure 7C).” on page 12, line 436. Again, according to figure 7C, there should be an increase, not lessened. There was no difference on CB1R and OXR2 co-localization between treatment groups in female mice.
Major concerns:
1. As mentioned before, there are mistakes in the text, the text and figure do not agree with each other.
2. It is not clear whether the cannabinoid-orexin receptor co-localization staining represent CB1R and OXR localizing in close proximity or they form heterodimer receptor, or both. The staining procedure is clearly written, however the quantification method is not.
3. The figures are too small, lines are too thin, and the text are illegible.
4. The anhedonic behavior showed no sex difference, however, there are sex differences in the cannabinoid-orexin receptor co-localization. The authors did not discuss the receptor co-localization changes according to the different function of these receptors in different nucleus. The physiological meaning of such change is not mentioned.
Author Response
Reviewer #1 (author responses in bold text)
This manuscript aimed to examine the cannabinoid-orexin receptor function in 24-h restrain stress-induced anhedonic behavior in male and female mice. The authors used sucrose preference test to access the anhedonic behavior. They first collected the baseline sucrose preference data. They then administered various cannabinoid and orexin receptor agonist or antagonists or combination of these agonist and antagonist to the mice right before the 24-h restrain stress. After the restrain stress, the mice were given the second drug injection then enter the final sucrose preference test. Afterwards the mice were euthanized and the blood and brain samples were collected. Blood samples were used to quantify plasma corticosterone levels. Brain samples were used in immunohistochemical staining experiments to localize the cannabinoid receptor type 1 and orexin receptor type 1 or orexin receptor type 2. The results indicated that YNT-185, the OXR2 agonist and the drug combination of CB1R antagonist SR141716A and TSC-1102 (here is a mistake in the text on page 6, line 283; it should be YNT-185) improved sucrose preference in stressed mice. There is no significant difference in body weight and plasma corticosterone levels in different treatment groups. Cannabinoid-orexin receptor co-localization study in male mice showed that there was increased CB1R and OXR1 co-localization in VTA and NAc of SR131716A+YNT-185 treated groups. In the mPFC, restrain stress increased the CB1R and OXR1 co-localization in vehicle treated group, and only the non-restraint mice showed increased co-localization by SR131716A+YNT-185 treatment. As for the co-localization of CB1R and OXR2, the SR132716A + YNT-185 treatment group showed significantly more co-localization than the SR141716A treatment group in the VAT. In the NAc, SR131716A+YNT-185 treatment reduced the CB1R and OXR2 co-localization in the non-restrained group. Restraint stress increased CB1R and OXR2 co-localization in YNT-185 treated group compared to non-restrain YNT-185 treated mice (However, the text said reduce, Page 12, line 421). There is no difference in mPFC. In the female mice, SR131716A+YNT-185 treatment significantly increased CB1R and OXR1 co-localization in the VTA. The authors wrote “Additionally, within this combination group, 24-h RST decreased CB1R-OX1R colocalization (p< 0.05) while no other drug treatments caused such a pattern (Figure 7A).” on page 12, line 432. However, according to Figure 7A, the restraint group showed increased co-localization, not decreased! Also, the authors wrote that “In the MPFC, 24-h RST lessened CB1R-OX1R colocalization in the 3mg/kg SR131716A + 40 mg/kg YNT-185 group (p< 0.001)(Figure 7C).” on page 12, line 436. Again, according to figure 7C, there should be an increase, not lessened. There was no difference on CB1R and OXR2 co-localization between treatment groups in female mice.
We thank the Reviewer for their detailed review of our Results section. These corrections have now been applied to the current version of the manuscript.
Major concerns:
- As mentioned before, there are mistakes in the text, the text and figure do not agree with each other.
Please refer to our first response.
- It is not clear whether the cannabinoid-orexin receptor co-localization staining represent CB1R and OXR localizing in close proximity or they form heterodimer receptor, or both. The staining procedure is clearly written, however the quantification method is not.
Section 2.5 has now been edited to provide additional details on our method of quantifying receptor colocalization. Finally, Section 3.4.1 – and elsewhere in the revised manuscript – now clarifies that receptor colocalization does not equate to heterodimerization, but is instead indicative of receptors existing in close proximity.
- The figures are too small, lines are too thin, and the text are illegible.
We agree with the Reviewer that many aspects of the figures were too small and difficult to read. We have increased the text and symbol sizes, as well as the line thickness of all graphs included in Figures 2–8. These have also been provided to the journal as separate files so that they are easier to read and access compared to these figures being embedded into the manuscript text.
- The anhedonic behavior showed no sex difference, however, there are sex differences in the cannabinoid-orexin receptor co-localization. The authors did not discuss the receptor co-localization changes according to the different function of these receptors in different nucleus. The physiological meaning of such change is not mentioned.
The third-last paragraph of the Discussion was edited to include new references and syntheses of data to explain how sex affects brain-region specific expression and function of cannabinoid and orexin receptors. This paragraph also ends with suggestions for future experiments that may address the knowledge gaps revealed by our results.
Reviewer 2 Report
This is a well-structured research article. The main question addressed by this research is the cannabinoid and orexin regulation of anhedonic behaviour caused by prolonged restraint stress.
This paper adds to this scientific area, as it attempts to expand current knowledge regarding cannabinoid-orexin receptor functionality in neuropsychiatric behaviours.
The introduction gives the background of this study as it briefly describes anhedonia and the main neuromodulatory systems involved.
“Materials and Methods” section is descriptive enough. It refers to the animals used, the techniques for the sample collection and the laboratory analysis as well as the statistical analysis that was performed.
The results are very interesting and, to my opinion, well presented.
The discussion is well written, summarizing and discussing the main findings of the study. The existence of a paragraph summarizing its main limitations, also adds to the scientific value of the paper.
Conclusions are consistent with the evidence presented. Perhaps the authors could present some specific targets for future studies.
References are relative to the subject and sufficient in number.
English language and style are generally fine. Minor issues need to be addressed before publication (e.g. in line 239, “was” should be replaced by “were” and in line 599, “signalling” should be replaced by “signaling”).
Author Response
Reviewer #2 (author responses in bold)
This is a well-structured research article. The main question addressed by this research is the cannabinoid and orexin regulation of anhedonic behaviour caused by prolonged restraint stress.
We thank the Reviewer for their positive feedback regarding our article’s research question and overall structure.
This paper adds to this scientific area, as it attempts to expand current knowledge regarding cannabinoid-orexin receptor functionality in neuropsychiatric behaviours.
The introduction gives the background of this study as it briefly describes anhedonia and the main neuromodulatory systems involved.
“Materials and Methods” section is descriptive enough. It refers to the animals used, the techniques for the sample collection and the laboratory analysis as well as the statistical analysis that was performed.
The results are very interesting and, to my opinion, well presented.
We appreciate the Reviewer’s interest and thought process as they navigated the article.
The discussion is well written, summarizing and discussing the main findings of the study. The existence of a paragraph summarizing its main limitations, also adds to the scientific value of the paper.
Conclusions are consistent with the evidence presented. Perhaps the authors could present some specific targets for future studies.
The second- and third-last paragraphs of the Discussion now include two future targets or directions that this research can take. The first idea proposes in vitro techniques, specifically the supplementation of sex hormones to cell cultures known to contain cannabinoid and orexin heterodimers. The second suggestion entails electrophysiological methods to elucidate the synaptic role of endocannabinoids within orexin circuitry in the striatum. These future studies may yield interesting results that will push this research topic forward.
References are relative to the subject and sufficient in number.
English language and style are generally fine. Minor issues need to be addressed before publication (e.g. in line 239, “was” should be replaced by “were” and in line 599, “signalling” should be replaced by “signaling”).
We thank the Reviewer for catching these mistakes. We have now corrected this language and spelling to improve readability.
Round 2
